# PA-LoFTR: Local Feature Matching with 3D Position-Aware Transformer

## Abstract

We propose a novel image feature matching method that utilizes 3D position information to augment feature representation with a deep neural network. The proposed method introduces 3D position embedding to a state-of-the-art feature matcher, LoFTR, and achieves more promising performance. Following the coarse-to-fine matching pipeline of LoFTR, we construct a Transformer-based neural network that generates dense pixel-wise matches. Instead of using 2D position embeddings for transformer, the proposed method generates 3D position embeddings that can precisely capture position correspondence of matches between images. Importantly, in order to guide neural network to learn 3D space information, we augment features with depth information generated by a depth predictor. In this way, our method, PA-LoFTR, can generate 3D position-aware local feature descriptors with Transformer. We experiment on indoor datasets, and results show that PA-LoFTR improves the performance of feature matching compared to state-of-the-art methods.

## 1 Introduction

Finding feature matching between images is an important task to many computer vision works, including camera calibration, structure from motion (SfM), visual localization, simultaneous localization and mapping (SLAM), stereo matching, etc. Generally, local feature matching problem can be solved with three stages: feature detection, feature description and feature matching. Most of existing methods follow the pipeline in sequence to handle feature matching. Feature detector narrows the focus to a set of interest points on images. Feature description phase generates corresponding descriptor for each interest point. Finally, correspondences between interest points of images are found by feature matching algorithms. While current methods try to extract features based on visual information from images, they pay little attention to the 3D position information of features, which limits the performance of matching.

As the development of deep learning, a number of works have introduced deep architecture to image feature matching. Some methods develop deep networks for feature detection and some focus on feature description or matching with learning process. Recently, several works have developed detector-free deep architectures that can generate dense matches (Rocco et al. (2018); Li et al. (2020); Rocco et al. (2020)). Detector-free methods can provide pixel-wise matches where feature detector can be dropped as each pixel is regarded as a potential interest point. Several detector-free methods achieve better performance for images with poor texture, complex patterns and illumination or large viewpoint change, where feature detector cannot determine enough interest points following the classic pipeline. Some methods, such as LoFTR (Sun et al., 2021) and CoTR (Jiang et al., 2021), introduce Transformer into image feature matching, where deep architectures can have global receptive field. Taking advantage of the ability to relate two features anywhere from the image, the methods can achieve state-of-the-art performance. However, the detector-free methods that generate dense matches can still give false correspondences under challenging pose changes or repetitive patterns. Though the Transformer-based architectures can learn global relation between features, current methods are still working with local visual features extracted directly from images, and most of them pay little attention to the 3D space information of feature points.

Based on observations above, we propose 3D Position-Aware Local Feature Transformer (PA-LoFTR), a novel approach that encodes space information for local feature matching. We follow

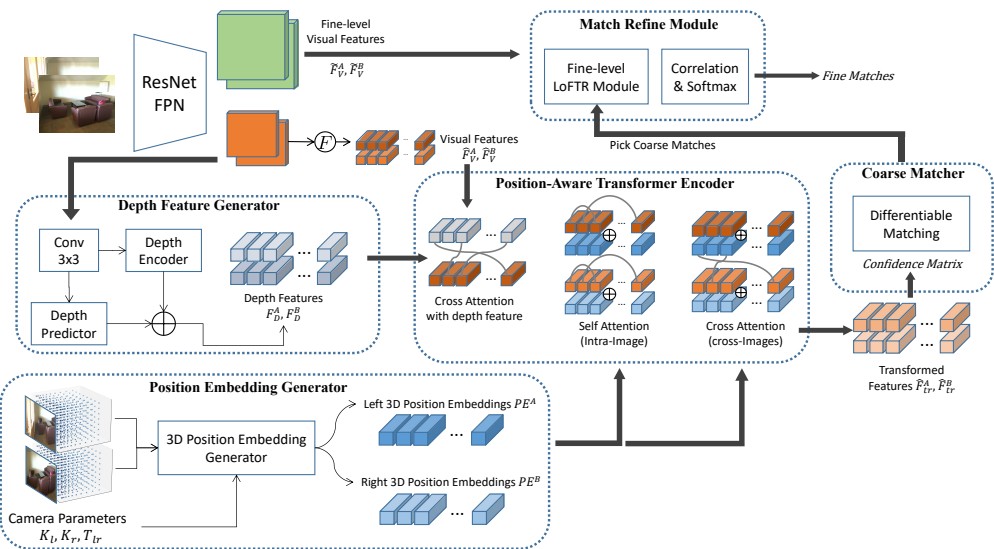

Figure 1: **Overview of Proposed method**. Given a pair of images $I_A, I_B$, PA-LoFTR uses a CNN backbone to give multi-level feature maps. Coarse-level feature maps $\hat{F}_V^A, \hat{F}_V^B$ are augmented by three modules: **1.** Depth Feature Generator predicts dense depth map for $\hat{F}_V^A, \hat{F}_V^B$, and gives corresponding depth features $F_D^A, F_D^B$. **2.** Position Embedding Generator prepares 3D position embeddings $PE^A, PE^B$ given camera intrinsic and extrinsic. **3.** Position-Aware Transformer Encoder combines 3D position embeddings and depth features into visual features $\hat{F}_V^A, \hat{F}_V^B$ with self and cross attention layers.

the coarse-to-fine structure of LoFTR to construct deep architecture. Inspired by the usage of position encoding for Transformer, we develop a 3D position embedding generator which encodes 3D point clouds for each pixel instead of encoding 2D pixel coordinates. When self and cross attention layers learn relations between local visual features, we add 3D position embeddings at each encoder layer in Transformer to boost learning process. Additionally, we construct a depth predictor branch to learn depth distribution of images, which can further help model learn space information. By encoding 3D position information and co-relating with depth features, PA-LoFTR learns meaningful features containing both visual and 3D space information that greatly help determine precise image matches. We evaluate the proposed architecture with a indoor dataset on camera pose estimation and test effectiveness of 3D position embedding on stereo matching tasks. The experiments show that PA-LoFTR can provide high-quality matches under challenging scenarios, and achieve state-of-the-art performance on some tasks. In this study, our main technique contributions are:

- We show that a 3D position encoder serving as position embedding generator for Transformer can greatly improve image correspondence quality determined by neural network.

- We propose a depth feature generator that give rough depth distributions for single image, which can help Transformer learn more space information and improve matching performance.

- We demonstrate that PA-LoFTR achieves state-of-the-art performance on indoor dataset and multiple tasks.

## 2    RELATED WORKS

**Classic Feature Matching Pipeline.** Classic feature matching pipeline includes feature detection, feature description and feature matching (Low (2004); Bian et al. (2017); Sattler et al. (2009); Tuytelaars & Gool (2000)). Many methods follow the classic feature matching pipeline to handle the image correspondence problem. SIFT (Lowe, 2004) introduces a way to build keypoint detector and

descriptor. ORB (Rublee et al., 2011) improves the efficiency to produce descriptors. Given detected features and their descriptors, RANSAC can filter outliers to provide precise matches (Fischler & Bolles, 1987). Classic methods have proved their effectiveness and have been widely applied in the industry.

**Feature Matching with Deep Learning Methods.** As the deep learning methods spring up, the performance of feature matching has been significantly improved under challenging scenarios, such as large change of viewpoint and illumination. Several methods introduce deep architectures to improve one or more stages of classic feature matching pipeline. LIFT (Yi et al., 2016) constructs a CNNs-based network to build feature detector and feature descriptor. MagicPoint (Detone et al., 2017) makes extra efforts to predict the corners in images. They have been proved effective in detecting keypoints with meaningful descriptors. SuperPoint (DeTone et al., 2018) presents a self-supervised framework to learn interest point detectors and descriptors based on MagicPoint, and SuperPoint applies homographic adaptation to improve the performance of feature detection. Methods based on classic pipeline use metrics to optimize matches, such as nearest neighbor to search the matches. SuperGlue (Sarlin et al., 2020) introduces an attentional graph neural network to match the local features captured by feature detector and descriptor. Several methods accomplish feature matching tasks based on deep learning methods (Efe et al. (2021); Rocco et al. (2022); Zhou et al. (2021); Truong et al. (2020); Dusmanu et al. (2019)).

Recently, some researchers have introduced Transformer to image correspondence problem. COTR (Jiang et al., 2021) adopts the architecture of DETR (Carion et al., 2020) that adopts Transformer encoder and decoder, and it tackles image correspondence problem in the way of querying pixel positions. LoFTR (Sun et al., 2021) regards each pixel in the image as a potential keypoint and apply Transformer encoder to learn relative relations between features, and it builds a coarse-to-fine pipeline to give precise matches. STTR (Li et al., 2021) conducts experiments that show Transformer can also be applied to stereo matching. The Transformer-based methods for image matching have shown the advantage of the global receptive field of Transformer, which can learn relations between features anywhere from the images. LoFTR can produce high-quality matches in challenging image pairs with low-texture areas and repetitive patterns. However, the usage of 2D or 1D position embeddings in methods above has a drawback, where position embeddings can not represent space correspondences between images. In this study, we propose a method that improves the performance of LoFTR by introducing meaningful 3D position embedding.

**Position Embedding in Transformer.** Recently, Transformer-based methods have been introduced to tackle multiple computer vision tasks, including object detection (Carion et al. (2020); Liu et al. (2022); Li et al. (2022); Jain et al. (2022); Zhang et al. (2022)), semantic segmentation (Miao et al. (2022); Li et al. (2022); Qian et al. (2022)), image feature matching (Jiang et al. (2021); Hong et al. (2022); Mao et al. (2022)), etc. While CNNs have the ability to aggregate features in a local window, Transformer can have global receptive field with attention mechanism that builds relations between features anywhere from the input sequences.

However, Transformer regards the input sequence as a set of unordered features, which leads to missing important position related information. To make use of the position information, transformer introduces position embeddings to the input features, which significantly improve the performance of models. However, when it comes to image matching problem, the widely used 2D position embedding methods (Carion et al. (2020); Jiang et al. (2021)) may serve as drawbacks. There are several methods working on improving the effectiveness of position embedding. STTR (Li et al., 2021) proves that adding position embeddings to the features will introduce a position-depended term in attention calculation, which is unrelated to image content, and STTR applies relative position embedding to relieve the problem. Yifan et al. (2022) tried to encode camera intrinsic and extrinsic to make better depth prediction. PETR (Liu et al., 2022) introduces 3D position embeddings to enhance DETR3D (Wang et al., 2021). PETR transforms 3D world space to camera frustum space, and embeds 3D coordinates by a multi-layer perception to produce 3D position embeddings, which further generates 3D position-aware image features. Nevertheless, recent Transformer-based methods for image matching still pay little attention to constructing position embeddings related to 3D space. Our work introduces 3D position embeddings to Transformer encoder to generate 3D position-aware image features for better performance of matching.

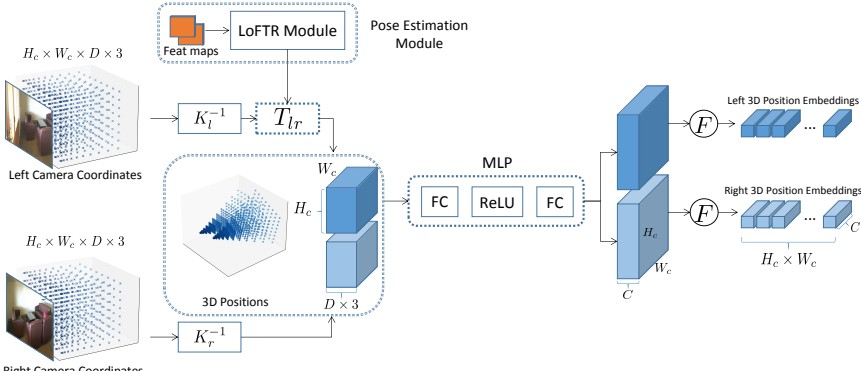

Figure 2: **Illustration of proposed 3D Position Embedding Generator.** For the pair of images given, a sparse 3D coordinate cloud is generated. Each coordinate represents a position at which a pixel can locate. Through camera intrinsic $K_l, K_r$ and extrinsic $T_{lr}$, the coordinate clouds are aligned under the right camera coordinate. Then we use multi-layer perception to transform the coordinate clouds to 3D position embeddings. Before inputting to Transformer Encoder, a flatten operation (F) is used to get position embedding sequences.

## 3 METHODS

### 3.1 OVERVIEW OF ARCHITECTURE

Figure 1 gives the overall pipeline of proposed PA-LoFTR. Given a pair of input images $I_A, I_B$, the backbone network ResNet FPN (Lin et al., 2017a) extracts multi-level visual features. We mainly concern the feature maps at $1/8$ and $1/2$ of original image shape, and we follow LoFTR (Sun et al., 2021) to denote them as coarse-level visual features $\hat{F}_V^A, \hat{F}_V^B$ and fine-level visual features $\tilde{F}_V^A, \tilde{F}_V^B$. We follow the coarse-to-fine structure of LoFTR, where coarse-level features are mainly used to generate matches which are refined with fine-level features.

Our proposed method focuses on coarse-level matching and improves performance by augmenting $\hat{F}_V^A, \hat{F}_V^B$ with Position-Aware Transformer Encoder. Position-Aware Transformer accepts visual features $\hat{F}_V^A, \hat{F}_V^B$ from backbone, depth features $F_D^A, F_D^B$ from depth feature generator, and 3D position embeddings as inputs, and outputs transformed features $F_{tr}^A, F_{tr}^B$. The transformed features are then used to calculate pair-wise similarity and keep coarse-level matches with high confidence. Finally, following LoFTR, the network looks into fine-level features around each pair of matches to refine final match positions.

### 3.2 3D POSITION EMBEDDING GENERATOR

Instead of using position embeddings from 2D pixel coordinates, we build the relation between 2D images and 3D space positions as shown in Figure 2. We first create meshgrid coordinates at the size of coarse-level feature maps, which is $H_c \times W_c$, by discretizing camera frustum space following DSGN (Chen et al., 2020) and PETR (Liu et al., 2022). We divide the depth space following linear-increasing discretization into $D$ positions and get a meshgrid of size $(H_c, W_c, D)$ where each cell represents a 3D position. The depth space is limited within $[d_{min}, d_{max}]$ along the center axis of camera and $D$ depth values are sampled, where each depth $d_i$ can be calculated as

$$d_{i,i\in\{0,1...,D-1\}} = d_{min} + \delta_d \times i \times (i+1), \text{where } \delta_d = \frac{d_{max} - d_{min}}{D(D+1)}. \quad (1)$$

Each 3D coordinate in the meshgrid $P$ with pixel $(u, v)$ on the image can be represented as $\mathbf{p}_i = (u \times d_i, v \times d_i, d_i)^T$. Further, as the image pair should share the same world space, the 3D coordinates are projected with camera intrinsic $K_l, K_r$ and extrinsic $T_{lr}$ where coordinate meshgrid of left image $P_{left}$ is aligned with meshgrid of right image $P_{right}$. The transformation to world space for a point follows Eq. 2, where camera extrinsic $T_{lr}$ is divided into rotation $R_{lr}$ and

translation $\boldsymbol{t}_{lr}$. Similar to PETR, we further normalize the 3D coordinates to a region of interest $[x_{min}, x_{max}, y_{min}, y_{max}, z_{min}, z_{max}]$, which depends on the dataset.

$$\boldsymbol{p}_l^{3d} = \boldsymbol{R}_{lr}\boldsymbol{K}_l^{-1}\boldsymbol{p}_l + \boldsymbol{t}_{lr}, \text{ where } \boldsymbol{p}_l \in \boldsymbol{P}_{left}$$
$$\boldsymbol{p}_r^{3d} = \boldsymbol{K}_r^{-1}\boldsymbol{p}_r, \text{ where } \boldsymbol{p}_r \in \boldsymbol{P}_{right} \tag{2}$$

After transformation to aligned world coordinates, the point cloud matrix $\boldsymbol{P}^{3d} \in \mathbb{R}^{H_c \times W_c \times (D \times 3)}$ is got for each image. Given the point cloud as input, a multi-layer perception (MLP) network generates 3D position embedding (PE) which is flatten to a sequence $\boldsymbol{PE} \in \mathbb{R}^{(H_c \times W_c) \times C}$, where $C$ is channel dimension. The 3D PE of each image is then input to Position-Aware Transformer Encoder which is described in section 3.4. As camera extrinsic is sometimes hard to get, we propose a pose refine branch where rough extrinsic is first predicted by LoFTR module. In more and more tasks, extrinsic can be available such as fix stereo camera or mechanical arm equipped with camera, we expect our method will be more effective in cases camera extrinsic is available.

### 3.3 DEPTH FEATURE GENERATOR

The purpose of Depth Feature Generator is to provide additional global scene information during learning of Position-Aware Transformer Encoder. Though predicting depth information from single image is also a challenging problem, we find it may guide network to learn better image correspondence. We propose a depth prediction branch after the backbone network as shown in Figure 3. Given the coarse-level feature map $\hat{F_V}$ of a single image, Depth Feature Generator predicts a dense depth distribution and outputs depth features for each pixel of the feature map.

Coarse-level feature is first going through a convolutional layer. The output features then walk through two branches where a Depth Encoder extract depth features $f_D$ with a single layer of Transformer encoder, and a Depth Predictor explicitly generates a dense depth distribution map. The depth encoder applies self-attention mechanism of Transformer to learn depth features. The dense depth map is used to generate depth embeddings similar to the way in MonoDETR (Zhang et al., 2022) and MonoDTR (Huang et al., 2022). To supervise learning of depth features, we follow the similar way in 3D Position Embedding Generator where depth space is discretized with LID into $k_{bins} + 1$ bins. The first $k_{bins}$ represents the foreground depth values and the last one is for background. The category of a ground-truth depth label can be calculated as

$$k = -0.5 + \sqrt{0.25 + \frac{d - d_{min}}{\delta_d}}, \text{ where } \delta_d = \frac{d_{max} - d_{min}}{k_{bins}(k_{bins} + 1)}. \tag{3}$$

We use a $1 \times 1$ convolutional layer to predict the depth category at each pixel and get a depth value through summation of results of softmax operation. We apply Focal loss (Lin et al., 2017b) to supervise category of depth for each pixel. Given the dense depth map $d_{map}$, we further interpolate the depth value among a learnable embedding pool to give depth embeddings $P_D$. The final depth features are generated as $\boldsymbol{F}_D = \boldsymbol{f}_D + \boldsymbol{P}_D$ where $\boldsymbol{F}_D \in \mathbb{R}^{(H_c \times W_c) \times C}$.

### 3.4 POSITION-AWARE TRANSFORMER ENCODER

After 3D position embedding generator and depth feature generator, Position-Aware Transformer Encoder aggregates all the information to give transformed features for matching. With position-aware feature embedding, the transformed features $\boldsymbol{F}_{tr}^A, \boldsymbol{F}_{tr}^B$ are more easily to match.

Here we introduce the basic idea of Transformer (Vaswani et al., 2017) as background. A Transformer encoder can contain a sequence of Transformer encoder layers, and the core component of a Transformer is the attention layer. The attention layer accepts three types of vectors as input, which are conventionally named as query, key, value. The key idea is that query vector $\boldsymbol{Q}$ will extract information from value vectors $\boldsymbol{V}$ based on the weights got from dot product with key vectors $\boldsymbol{K}$ where each value vector has its corresponding key vector. The dot product serves as a similarity score between query and keys which enables query $\boldsymbol{Q}$ to select more relevant values. The formal equation of attention layer computes as

$$Attn(\boldsymbol{Q}, \boldsymbol{K}, \boldsymbol{V}) = \text{softmax}(\boldsymbol{Q}\boldsymbol{K}^T)\boldsymbol{V}. \tag{4}$$

Many studies have made efforts in designing effective and efficient attention layers including multi-scale deformable attention (Zhu et al., 2020), Quad-Tree Attention (Tang et al., 2022), etc. In our

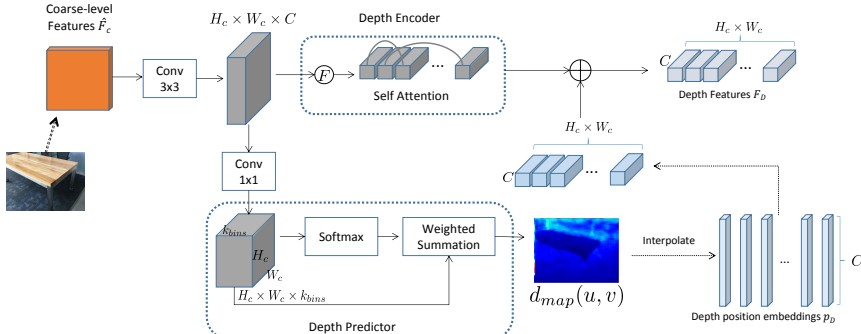

Figure 3: **Depth Feature Generator in PA-LoFTR**. Depth Feature Generator provides depth feature for each image of image pairs. Coarse-level visual feature $\hat{\boldsymbol{F}}_V$ is first transformed by a convolutional layer, and go into two branches. **1.** Depth predictor gives a dense depth map $\boldsymbol{d}_{map}$ over the feature map, and generates depth embeddings $\boldsymbol{p}_D$ for each pixel. **2.** Depth Encoder apply single Transformer encoder layer and combine depth embeddings to generate depth features $\boldsymbol{F}_D$.

work, we follow LoFTR and apply Linear Attention (Katharopoulos et al., 2020) in the Transformer Encoder, which reduces the computation cost of Transformer to $O(N)$ with the length of input sequences.

In our proposed Position-Aware Transformer Encoder, each Transformer Encoder layer contains three stages as shown in Figure 1. The first cross attention layer accepts visual features $\hat{\boldsymbol{F}}_V^i$ as queries and depth features $\boldsymbol{F}_D^i$ as keys and values, and here $i \in \{A, B\}$ means the first or second of image pair. The middle self attention layer builds relation for features across a single image, where it accepts $\hat{\boldsymbol{F}}^i$ ($\hat{\boldsymbol{F}}_A$ or $\hat{\boldsymbol{F}}_B$ out of the first cross attention layer) as queries, keys and values. The last cross attention layer relates features between the pair of images, where $\hat{\boldsymbol{F}}_A$ or $\hat{\boldsymbol{F}}_B$ serves as queries and $\hat{\boldsymbol{F}}_B$ or $\hat{\boldsymbol{F}}_A$ as keys and values. Most importantly, position embeddings are vital to learning of Transformer Encoder. Different from LoFTR where 2D PE is used once at the beginning of Transformer Encoder, we apply 3D PE to the middle self attention and the last cross attention in each of the Transformer Encoder layers.

After feature Transformer Encoder, the transformed features $\hat{\boldsymbol{F}}_{tr}^A, \hat{\boldsymbol{F}}_{tr}^B$ are input to coarse-level matching module to select confident matches. A score matrix is calculated between two set of features where $S(i, j) = \frac{1}{\tau} \cdot \langle \hat{\boldsymbol{F}}_{tr}^A(i), \hat{\boldsymbol{F}}_{tr}^B(j) \rangle$, and we follow LoFTR to apply dual-softmax operator (Sun et al., 2021) to $S$ to obtain mutual nearest neighbor matching by the matching probability $P_c(i, j) = \text{softmax}(\boldsymbol{S}(i, \cdot))_j \cdot \text{softmax}(\boldsymbol{S}(\cdot, j))_i$. With coarse matches selected, coarse-to-fine module applies spatial expectation to each pair of coarse matches to get final precise matches.

### 3.5 IMPLEMENTATION DETAILS

Following LoFTR, we trained PA-LoFTR on the indoor dataset ScanNet (Dai et al., 2017). As mentioned in LoFTR, the original LoFTR model is trained with 64 GTX 1080Ti GPUs with full ScanNet dataset containing 1513 scenes, which requires too much resources and time cost for researchers. In this study, we apply a lite training setting, where we split partial of ScanNet which contains 100 scenes for training with 4 GTX 1080Ti GPUs and a batch size of 8, and test on the data split following Sarlin et al. (2020). To boost training, we use the pretrained CNN backbone from LoFTR, and the model parameters left in the following network are initialized with random weights, and the dimension of features is set as 128 instead of 256 to save memory. We further test model on KITTI Stereo dataset (Geiger et al., 2013), where we fine-tune the PA-LoFTR trained on ScanNet. As we notice that in many tasks, camera extrinsic is hard to get, which is a limit of our model, we implement a pose-refine version of PA-LoFTR with a simplified LoFTR branch to estimate camera extrinsic.

| Method | KITTI 2015 | |
|---|---|---|
| | EPE (px) | 3px Error (%) |
| PSMNet (Chang & Chen, 2018) | 6.56 | 27.79 |
| GWC-Net-g (Guo et al., 2019) | 2.21 | 12.60 |
| AANet (Xu & Zhang, 2020) | 1.99 | 12.42 |
| STTR (Li et al., 2021) | 1.50 | 6.74 |
| STTR (with our 3D PE generator) | 1.55 | 5.75 |
| LoFTR† | 1.14 | 5.32 |
| PA-LoFTR (ours) | **1.04** | **5.19** |

Table 1: **Stereo Matching Results.** Models are tested on KITTI Stereo 2015 training split, following STTR. PA-LoFTR can give sparse disparity of high quality. We test STTR model with 3D PE generator, and the results can also reach state-of-the-art performance.

| Method | ScanNet | |
|---|---|---|
| | precision (%) | recall (%) |
| LoFTR† | 81.58 | 34.12 |
| LoFTR | 87.87 | **57.05** |
| PA-LoFTR (ours) | **93.74** | 49.39 |

Table 2: **Precision and Recall.** Detector-free Models are tested on ScanNet testing split. Though trained with smaller dataset, PA-LoFTR can achieve better precision compared with LoFTR.

## 4 EXPERIMENTS

We experiment our method on multiple tasks, including precision and recall, stereo disparity and relative pose estimation. For pose estimation, we propose a pose-refine structure where camera extrinsic is first estimated and refined by PA-LoFTR to get the final pose. PA-LoFTR shows that it can get matches of high quality with the help of 3D position embedding generator.

**Stereo Matching.** We choose KITTI dataset to evaluate PA-LoFTR and the 3D Position Embedding Generator. We trained a PA-LoFTR model on ScanNet following our lite training setting, and we fine-tune the model on KITTI dataset. For fair comparison, we do not tune model on KITTI Stereo dataset. Instead we fine-tune the model on KITTI object detection dataset with pseudo ground truth disparity following disp-RCNN (Sun et al., 2020). Though the model structure of PA-LoFTR is not designed for stereo matching where matches only occur on horizontal axis of images, we find PA-LoFTR can directly generate matches along horizontal lines with high quality. Compared to state-of-the-art methods targeting at disparity estimation, PA-LoFTR can provide disparity for the detected matches with high accuracy. We also apply 3D PE Generator to state-of-the-art model STTR (Li et al., 2021) targeting at stereo matching to show effectiveness, where model structure is similar to ours. We fine-tuned STTR with 3D PE Generator on KITTI Stereo dataset and report metrics on validation split. The results are shown in Table 1.

**Match Evaluation.** We compared PA-LoFTR with the detector-free method, LoFTR, on indoor dataset, ScanNet. As shown in Table 2, precision and recall are reported where PA-LoFTR can generally give matches with higher accuracy. Besides, The accuracy of camera pose estimated from the corresponding points is typically used to evaluate the quality of feature correspondences. The evaluation protocol follows SuperGlue (Sarlin et al., 2020), where feature matches generated are used to solve essential matrix with RANSAC. We report the AUC of pose error at thresholds of $5°, 10°, 20°$, and pose error is defined as the maximum of angular error in rotation and translation following Sarlin et al. (2020). Camera extrinsic is used to construct 3D Position Embedding in PA-LoFTR which greatly improve the quality of feature matching. In relative pose estimation task, camera extrinsic is unavailable and we propose a pose-refine mechanism to fit PA-LoFTR. We add a pre-trained LoFTR branch that generates a pose estimation first, and then PA-LoFTR constructs the 3D position embedding to get final matches. The details of implementation can be found in supplementary materials. The results in Table 3 show that PA-LoFTR with pose-refine mechanism can improve the estimated pose by a large margin compared to LoFTR under the same training setting.

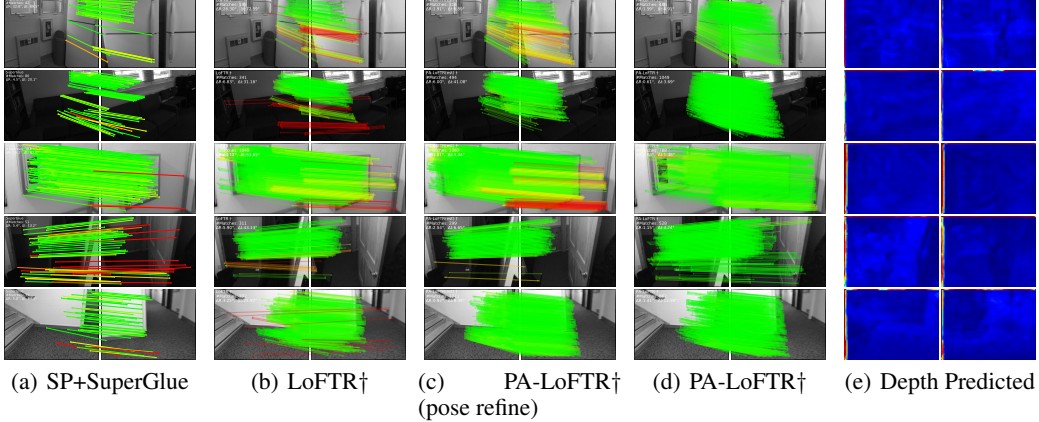

| (a) SP+SuperGlue | (b) LoFTR† | (c)  PA-LoFTR† (pose refine) | (d) PA-LoFTR† | (e) Depth Predicted |
|---|---|---|---|---|

Figure 4: **Qualitative Results.** PA-LoFTR is compared to LoFTR (Sun et al., 2021) and Super-Glue (Sarlin et al., 2020) on indoor dataset, and depth map predicted is visualized. The proposed pose-refine PA-LoFTR can provide more correct matches and less false matches compared to SuperGlue and LoFTR under challenging cases. PA-LoFTR can generate even better results where ground truth relative pose is provided to the model. The red color means epipolar error of the match is beyond $5 \times 10^{-4}$.

We conduct the experiments with the lite training setting as described in section 3.5 to show effectiveness of PA-LoFTR. As shown in Table 3, PA-LoFTR can improve the performance of pose estimation compared to LoFTR under the same training settings. It is noticed that out method can provide accurate matches on testing dataset even trained with limited splits of ScanNet, which shows its ability of generalization. It is expected that PA-LoFTR can have even better performance if we can train the model following the setting of original LoFTR, where all 1513 scenes of ScanNet are used for training and 64 GPUs are used with a batch size of 64. We also test the case where

| Method | ScanNet | | |
|---|---|---|---|
| | auc@5° | auc@10° | auc@20° |
| ORB+GMS (Bian et al., 2017) | 5.21 | 13.65 | 25.36 |
| SP+SuperGlue (Sarlin et al., 2020) | 16.16 | 33.81 | 51.84 |
| LoFTR* (Sun et al., 2021)) | 20.06 | 40.8 | 57.62 |
| LoFTR† | 13.08 | 27.42 | 43.41 |
| PA-LoFTR† (pose refine) | 17.04 | 34.28 | 51.18 |
| PA-LoFTR† | **27.74** | **47.35** | **64.19** |

Table 3: **Relative Pose Estimation on ScanNet dataset.** The AUC of pose error is reported, which reflects quality of feature matching detected. PA-LoFTR can improve matching performance a lot. Without camera extrinsic, PA-LoFTR can also achieve promising results compared with state-of-the-art methods. ∗ means full training setting in LoFTR with 64 GPUs and a batch size of 64 on full ScanNet dataset. † means our lite training setting on partial ScanNet where 4 GPUs are used with a batch size of 8 as described in section 3.5. Models are tested on consistent dataset following SuperGlue (Sarlin et al., 2020).

ground truth pose is available, and PA-LoFTR can show its full power and give high-quality feature matches. We expect that in tasks where camera extrinsic is available, 3D position embedding can provide stronger prior knowledge to the model and PA-LoFTR can show more effectiveness.

**Ablation Study.** Compared to LoFTR, the main module is the 3D Position Embedding Generator which bridges the gap between camera extrinsic and feature embedding for Transformer. We evaluate 4 variant settings of 3D Position Embedding Generator as in Table 4: 1) Apply only rotation matrix of camera intrinsic to generator position embeddings, which results in a drop of accuracy. 2) Apply position embeddings once at the beginning of Transformer Encoder. 3) Apply position embeddings for self-attention layers in Transformer Encoder. We find decreasing the use of 3D position

| Method | ScanNet | | |
|---|---|---|---|
| | auc@5° | auc@10° | auc@20° |
| **PA-LoFTR** | 27.74 | 47.35 | 64.19 |
| $R$ matrix for 3D PE | 23.21 | 42.98 | 58.67 |
| One-time position embedding | 24.91 | 43.40 | 59.87 |
| Position embedding for self-attention layers | 26.10 | 44.82 | 61.08 |
| Estimated pose for position embedding | 17.04 | 34.28 | 51.18 |
| Without Depth Feature Generator | 28.27 | 48.31 | 65.39 |
| Depth feature for 3D position embedding | 21.08 | 39.79 | 56.98 |
| Position embedding for the first cross attention | 27.16 | 46.69 | 63.56 |

Table 4: **Ablation Study.** Four variants of 3D PE Generator and three variants of usage of depth features by Depth Feature Generator. All models are trained under lite setting † as described in Table 3. All models, except for PA-LoFTR with estimated pose for 3D PE Generator, are trained with ground truth camera extrinsic. PA-LoFTR without Depth Feature Generator performs better on testing dataset, but has lower performance (3 points decrease) on validation dataset.

embedding in Transformer layers will both decrease accuracy. 4) Apply estimated pose with pretrained LoFTR to generator 3D postion embeddings instead of using ground truth extrinsic, which is essential for pose estimation task. The usage of estimated pose will cause decrease of matching performance compared to full PA-LoFTR, but can explicitly refine the estimated pose.

We also evaluate 3 variant settings related to usage of depth feature generator: 1) Do not apply depth feature generator. While it leads to decrease in performance on validation dataset compared to full PA-LoFTR (about 3 points decrease), the model performs better on testing dataset. The Depth Feature Generator may have limited ability of generalization. It is expected to have better performance if trained with more scenes. 2) Apply depth feature in 3D PE generator, where we use depth probability to guide generation of 3D PE, which does not give better performance. 3) Apply 3D position embedding at the first cross attention layer between depth features and visual features, which does not improve performance compared to giving not 3D PE to the first cross attention.

From the experiments, we can conclude that 3D position embeddings can have strong boost to image matching with Transformer. When the ground truth camera extrinsic is given, a strong prior scene information can be extracted from 3D PE Generator to help PA-LoFTR find accurate matches and less false matches. When ground truth camera extrinsic is not available, we estimate a raw camera relative pose with a pretrained LoFTR branch, and PA-LoFTR can still get enough scene information to generate better feature matches. Implementation details can be found in supplementary materials.

## 5 CONCLUSION

We presents a novel feature matching approach, PA-LoFTR, which introduces 3D position embeddings to Transformer encoder to bridge the gap between 3D position information and 2D image features. The proposed method embeds the points cloud for each image as position features and applies self and cross attention layers to transform image features with 3D space information. PA-LoFTR can detect precise and plenty of feature matches even under challenging pose changes. The experiments show that PA-LoFTR can reach state-of-the-art performance, and we expect PA-LoFTR can be extended to more tasks in which camera extrinsic is known, where PA-LoFTR can show its full power. Source code will be released soon.

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
