# OpenReview forum: "PA-LoFTR: Local Feature Matching with 3D Position-Aware Transformer"
_ICLR.cc/2023/Conference — Submitted to ICLR 2023_

### Official Review · Reviewer_kZ1E · 2022-10-21

**Confidence:** 5
**Correctness:** 2
**Technical Novelty And Significance:** 2
**Empirical Novelty And Significance:** Not applicable
**Recommendation:** 3

**Clarity, Quality, Novelty And Reproducibility:**

Clarity : Good;
Quality: Fair;
Novelty: Fair;
Reproducibility: Fair;

**Strength And Weaknesses:**

[Strength]

1.  Using 3D position embedding to replace the original 2D pixel position embedding is interesting.
2. Results on the KITTI Stereo 2015 validation results are good.
3. Overall, this paper is easy to follow.

[Weaknesses]
The main weaknesses of this paper are experiments.

1. Since the proposed 3D position embedding module requires a known relative pose, the proposed method has limited applications.

a) Applying the proposed method to the KITTI Stereo 2015 dataset is good. However, only results on the validation set are given. Please report the results on the testing set;
b) For the relative pose estimation task, though authors show that the proposed method can refine an estimated pose from Lofter, the performance gap between the proposed method and the original Lofter is big. To address this, I would expect:

i) Exactly following the training configuration of the original Lofter, and see whether the proposed 3D position embedding module still works, i.e., can refine an estimated pose from Lofter;

ii) Add a baseline experiment using the original 2D pixel position embedding and see whether the proposed 3D position embedding module can obtain better performance under the authors' setting;

iii) Using more recent networks (e.g., QTA) to check the proposed 3D position embedding module still works;

iv) Using the YFCC100M and IMC2021 datasets to check whether the proposed method can work on outdoor datasets.

c) For the ablation study, authors use the GT relative pose. It will make this section questionable;

d) Please add an ablation study, showing the robustness of the proposed method with respect to prior relative poses at a different level of accuracy.

2. [Minor] typos. For example,

a) Gnerator in Figure 1,
b) Self Attention (Inter-Image) ----->>>> Self Attention (Intra-Image)
c) Please bold matrix/vectors.




**Summary Of The Paper:**

This paper introduces a 3D position embedding module to a state-of-the-art feature matcher, LoFTR.

Specifically, the authors develop a 3D position embedding generator that encodes 3D point clouds for each pixel instead of encoding 2D pixel coordinates.

The proposed 3D position embedding generator requires a known relative pose from GT or from a third-party relative pose estimator.

Experiments on the Scannet and Kitti datasets show the effectiveness of the proposed method.

**Summary Of The Review:**

The main weakness of the proposed method is relying on a known relative pose, either from GT or from a third-party estimator.

The experiments fail to show that using a third-party estimator (Lofter) would make the proposed method outperforms the third-party estimator.

Please use the same training setting as the third-party estimator.

---

> ### Author Response · Authors · 2022-11-17
> **Updates and about your concerns**
>
> Thank you for your valuable comments and advice on our work. We have made updates in the paper with additional experiments and explanation.
>
> * **Q1**: *Since the proposed 3D position embedding module requires a known relative pose, the proposed method has limited applications.*
>
> * **A1**: You are right that in some applications, our method might be less effective where pose information is not available. However, in many studys, pose information may become easier to get such as a mechanical robot arm is available or in fixed stereo cases. Our method may show that with extra pose information injected, what is the upper limit the model can achieve on image matching problem.
>
> * **Q2**: *Advice on further experiments.*
> * **A2**:
>   * As we mentioned in the paper, exactly following the original training setting as LoFTR has done is not available for us currently, and we may try to find a way in the future that can compare our method with LoFTR more fairly.
>   * Actually, LoFTR is a method that applies 2D pixel position embedding to handle image matching. In Table4, only equipping with 3d position embedding in our method can also achieve better results compared with LoFTR.
>   * Another point we want to make is that we tried to show the quality of matching results given by our method is better, and we tried to use pose estimation benchmark to compare with previous work. To make sense, we add an experiment that directly compare the precision and recall in Table 2.
>   * Thank you for your advice that experimenting more outdoor datasets to show robustness. We will conduct more experiments in the future work.

---

### Official Review · Reviewer_PA3N · 2022-10-24

**Confidence:** 2
**Clarity, Quality, Novelty And Reproducibility:** Please see summary
**Correctness:** 3
**Technical Novelty And Significance:** 2
**Empirical Novelty And Significance:** 2
**Recommendation:** 3

**Strength And Weaknesses:**

Please see summary

**Summary Of The Paper:**

This manuscript proposes a method, PA-LoFTE, for solving feature matching between images which generally follows a detection-description-matching three-stage pipeline. In order to achieve that, the authors utilize depth information from a depth predictor to generate 3D position embedding, then combine visual features, depth features, and 3D position embeddings with transformer.
The discussion on the necessity of 3D space information is limited, which is the major motivation of the proposed method. The main contribution is leveraging the 3D position embedding generator for feature generation. They perform an empirical study on indoor dataset and multiple tasks and conclude the superior performance of PA-LoFTR.

Overall, this paper could be a borderline paper. Given the clarification in the rebuttal, I would increase the score.

For the motivation:
1. It should be aware depth estimation the similar challenges to correspondence prediction, why depth estimation helps correspondence prediction?

Some suggestions for the experiments:
1. Compared with LoFTR, PA-LoFTR injects 3D position information into the original Coarse-Level Local Feature Transform. The proposed method (depth feature generator and 3D position embedding generator) is effective based on LoFTR, how about adopting other baselines?
2. In Table 1, how about the performance of LoFTR?
3. In Table 3, Without Depth Feature Generator performs better on the test dataset while having lower performance on the validation dataset, could you please discuss this problem?
4. Could you please give some internal estimated depth images in the depth feature generator?

Minor comments:
1. Figure 1: Depth Feature Gnerator -> Depth Feature Generator.

**Summary Of The Review:**

Overall, this paper could be a borderline paper. Given the clarification in the rebuttal, I would increase the score.

---

> ### Author Response · Authors · 2022-11-17
> **Updates and about your concerns**
>
> Thank you for your valuable comments on our work. We have made updates in the paper with additional experiments and explanation. The intermediate results of network have also been visualized in the figure.
>
> * **Q1**:
>   * *It should be aware depth estimation the similar challenges to correspondence prediction, why depth estimation helps correspondence prediction?*
>   * *In Table 3, Without Depth Feature Generator performs better on the test dataset while having lower performance on the validation dataset, could you please discuss this problem?*
> * **A1**: Yes you are right that depth prediction is also a challenge in 3D reconstruction problem, and it might be as hard as correspondence prediction. In our method, we want to introduce depth predictor is that we want to see whether the two branches can boost each other. As we may notice that to exactly and directly know correspondences between two images, we need relative pose information and depth information for each image. In our method, as we have injected pose information into the network, we want to see if depth predictor can also give a better performance to help determine image correspondences. The results may show that given a better predicted depth result, for example on validation dataset or similar scenes to training dataset, the model can generate even better matching results.
>
> * **Q2**: *Compared with LoFTR, PA-LoFTR injects 3D position information into the original Coarse-Level Local Feature Transform. The proposed method (depth feature generator and 3D position embedding generator) is effective based on LoFTR, how about adopting other baselines?*
> * **A2**: Thank you for your advice, and we think it is a good direction that we may test whether 3D position embedding can work in other scenarios. In the current work, we have tried to test 3D position embedding in STTR ( (Li et al., 2021)) model to see whether it is effective on disparity prediction. It turns out the effect might not be that obvious compared to matching between arbitary images. It might be the reason that poses between stereo images are fixed and 3D position embedding may provide limited prior knowledge. In the future work, we may try to experiment on other architectures.
>
> * **Q3**: *In Table 1, how about the performance of LoFTR?*
> * **A3**: We have conducted additional experiment for LoFTR, and you can see the results in Table 1 now.
>
> * **Q4**: *Could you please give some internal estimated depth images in the depth feature generator?*
> * **A4**: We have updated the intermediate depth prediction results in Figure 4.

---

### Official Review · Reviewer_iea1 · 2022-10-24

**Confidence:** 4
**Correctness:** 3
**Technical Novelty And Significance:** 2
**Empirical Novelty And Significance:** 2
**Recommendation:** 3

**Clarity, Quality, Novelty And Reproducibility:**

- The paper is written clearly with useful figures.
- The ablation study is especially good and informational, which clearly shows the strengths and weaknesses of the proposed method.
- It should be reproducible since it’s based on an established approach with well tested data. The authors also mentioned the code will be published soon which should make reproducing the results easy.


**Strength And Weaknesses:**

Strengths:
- The paper is written well with many ablation studies and numerical findings. The results also show that the proposed 3d positional embedding helps.

- In the ablation study, the author illustrates the performance with and without the depth predicted, and presents a few failed approaches to make it work. Though the result probably would work against the proposed method in the paper, I’d appreciate this part which makes it clearer where it works better.

Weakness:
- The idea of 3d position embedding is not new. For example, this recent paper “Input-level Inductive Biases for 3D Reconstruction” described how to encode camera intrinsics and extrinsics into a transformer in positional encoding. As the paper mentions, PETR (Liu et al. 2022) also introduces a similar idea to the problem of object detection. Therefore, the application of a similar idea to the specific problem (using a transformer to do image matching) is not very novel.

- It’s a bit weird that the paper contributes a big chunk to depth predictors and in the end it doesn’t generalize well to many scenarios (Sec 4.3). In addition, training the depth predictor as a joint head would also make the training harder, converge slower, and increase the overhead for the model to be deployed. It’s realistic not to assume that the depth head trained on ScanNet would generalize well on KITTI. In my opinion, if you want to try to add depths, starting with a pretrained mono-depth model would be a better route. Since it doesn’t add overhead in the training process, and possibly would generalize better since it’s trained on a large corpus of data.


**Summary Of The Paper:**

This paper proposes a descriptor-free feature matching method based on LoFTR. Given a pair of input images, it uses a transformer architecture to calculate pairwise similarity between two sets of coarse features, and then optimize the fine-grained positions with fine-level features. The main differentiating factor compared with previous work is a 3d positional embedding generator that makes the transformer encoder aware of 3d information. A depth prediction head is also jointly learnt to provide depth encoding to aid the feature encoding. The proposed method achieves a large margin over the previous SoTA methods.


**Summary Of The Review:**

I recommend reject due to the following reasons:
- The proposed method should intuitively improve the results as a better positional embedding, incorporating more inductive bias such as 3d information of the input features has been demonstrated to improve the results quite well. However, the novelty is a bit hurt because this point has been shown on other domain-specific tasks. In addition, this paper is heavily based on LoFTR, the idea of using a transformer to do descriptor-free image matching.

- The emphasis on the depth predictor as a contribution is a bit problematic, as the results seem to show contradictory conclusions. I suggest using an off-the-shelf depth predictor to help with this argument.

---

> ### Author Response · Authors · 2022-11-17
> **Novelty and Contribution of Depth Predictor**
>
> Thank you for your valuable comments on our work. Here we give our updates and give response about your concerns.
>
> * **Q1**: *The idea of 3d position embedding is not new. For example, this recent paper “Input-level Inductive Biases for 3D Reconstruction” described how to encode camera intrinsics and extrinsics into a transformer in positional encoding. As the paper mentions, PETR (Liu et al. 2022) also introduces a similar idea to the problem of object detection. Therefore, the application of a similar idea to the specific problem (using a transformer to do image matching) is not very novel.*
> * **A1**: Thank you for your advice and we have included the mentioned paper in our related works. You are right that the paper mentioned proposed a method that encodes camera intrinsic and extrinsic into Perceiver network and predict depth information from multi-view images, but it seems that it cannot solve corresponding problem directly. However, we think applying 3d position embedding to an end-to-end deep learning architecture for image matching is still an interesting topic that deserves deep digging. Through encoding extra camera parameters into Transformer for image matching problem, we may find an upper limit of using Transformer models to handle the problem. Also for some applications where pose information is available, our method does improve the matching results greatly. We update the precision and recall rate compared with LoFTR in Table 2, and it turned out that PA-LoFTR can provide more accurate image correspondences.
>
> * **Q2**: *It’s a bit weird that the paper contributes a big chunk to depth predictors and in the end it doesn’t generalize well to many scenarios (Sec 4.3). In addition, training the depth predictor as a joint head would also make the training harder, converge slower, and increase the overhead for the model to be deployed. It’s realistic not to assume that the depth head trained on ScanNet would generalize well on KITTI. In my opinion, if you want to try to add depths, starting with a pretrained mono-depth model would be a better route. Since it doesn’t add overhead in the training process, and possibly would generalize better since it’s trained on a large corpus of data.*
> * **A2**:
>   * Truely, as described in our paper that we find that depth predictor can further improve matching results on validation dataset. Actually, many studys do have deep dive in predicting depth from single image. The reason we want to introduce depth predictor is that we want to check whether the two branches can boost each other where both problems are related with 3D reconstruction. The results may show that given a better predicted depth result, for example on validation dataset or similar scenes to training dataset, the model can generate even better matching results.
>   * Another point we want to make is that the model with depth generator do work well on KITTI dataset after tuning on KITTI object detection dataset as described in Experiment section. You are right that directly applying pretrained model on Scannet cannot generalize well on KITTI dataset, but in our experiments, it shows that model can still provide mostly correct matches given the image pair. It is more likely that the depth predictor just doesn't provide any useful depth information without tuning.
>   * And thank you for your advice that we may try to apply a pretrained off-the-shelf depth predictor to the current model in the future work. We may also update the network architecture and try to make two branches boost each other.

---

### Decision · Program_Chairs · 2023-01-20

**Decision:**

Reject

**Justification For Why Not Higher Score:**

This paper receives 3x rejects, the decision is clear.

**Justification For Why Not Lower Score:**

NA

**Metareview: Summary, Strengths And Weaknesses:**

This paper receives 3x rejects, the decision is clear.